

# Empirical Estimation of the Spatial Sediment Transport Capacity Coefficient Using the Rain Erosivity Factor and SWAT Model Results in the Han River Basin, South Korea

**Chung-Gil Jung[1], Won-Jin Jang[2], Seong-Joon Kim[3,*]**

[1]Dept. of Civil, Environmental and Plant Eng., Konkuk University, Seoul, 143-701, South Korea; wjd0823@konkuk.ac.kr
[2]Dept. of Civil, Environmental and Plant Eng., Konkuk University, Seoul, 143-701, South Korea; jangwj0511@konkuk.ac.kr
[3,*]Dept. of Civil, Environmental and Plant Eng., Konkuk University, Seoul, 143-701, South Korea; kimsj@konkuk.ac.kr

*Correspondence to*: kimsj@konkuk.ac.kr; Tel.: +82-02-444-0186

**Abstract.** The ratio of sediment delivery is a critical and uncertain factor in model-based assessments of the total sediment
yield of watersheds that results from the transport of sediment produced by soil erosion. This study estimates the watershed-
scale distribution of sediment yield at a spatial resolution of 1 km by 1 km through evaluating the rain erosivity (R) factor of
the Revised Universal Soil Loss Equation (RUSLE) in the Han River basin (34,148 km$^2$) of South Korea over 14 years
(2000~2013) using 1-minute data from 16 rainfall gauging stations. For this study, the Water and Tillage Erosion
Model/Sediment Delivery Model (WATEM/SEDEM) sediment delivery algorithm is adopted. This algorithm is based on R,
the soil erodibility factor K, the length-slope factors LS of RUSLE, and the transport capacity coefficient KTC. The average
1-minute value of R for the basin is estimated to be 3,812 MJ/ha·mm/year. To determine the 1-km grid-based transport capacity
coefficient (KTC; generally given from 0.01 to 100) for the transport capacity (TC) equation used in the estimation of sediment
transport with WATEM/SEDEM algorithm, the TC results from 181 subwatersheds ranging in area from 75.4 km$^2$ to 281.5
km$^2$ obtained using the Modified Universal Soil Loss Equation (MUSLE) implemented in the Soil Water Assessment Tool
(SWAT) are used. A comparison of the suspended solids (SS) simulated using SWAT with the observed values at 7 locations
yields an average coefficient of determination R$^2$ of 0.72. Using the SWAT TC, the spatial KTC is determined in each
subwatershed. These values range from 0.16 to 112.58, and the average value for the whole basin is 12.58. To permit general
estimation of KTC values, multiple regressions are performed using the characteristic watershed factors of watershed slope,
watershed area, the K factor of MUSLE, upland crop area (%), and paddy field area (%). A multiple regression equation of
KTC with watershed area, K factor, and upland crop area (%) is derived. This equation yields an R$^2$ of 0.76 when compared
to the KTC values evaluated using SWAT. The KTC can be determined using information on watershed scale, soil and land
use.





## 1 Introduction

Soil erosion and sedimentation by water involves the processes of detachment, transportation, and deposition of sediment by raindrop impact and flowing water (Foster and Meyer, 1977; Wischmeier and Smith, 1978; Julien, 1998). The major forces originate from raindrop impact and flowing water. The mechanisms of soil erosion take place where water from areas of sheet

flow runs together under certain conditions and forms small rills that produce small channels. When the flow becomes concentrated, it causes erosion, and much material can be transported within these small channels. A few soils are very susceptible to rill erosion. These rills gradually join together to form progressively larger channels, with the flow eventually proceeding to established streambeds. Some of this flow becomes great enough to create gullies. Soil erosion may not be noticeable on exposed soil surfaces, even where the raindrops are eroding large quantities of sediment; however, erosion can

be dramatic where concentrated flows create extensive rill and gully systems (Kim, 2006).

Extremely heavy rainfall events have increased over the past few decades due to climate change (IPCC 2007). Climate change results in changes in land cover type, biomass, and hydrologic regimes and subsequently affects erosion on hillslopes. The removal of topsoil reduces the productivity of land, and sediment-bound nutrients increase the growth and proliferation of aquatic organisms such as algae, thus producing off-farm impacts (Pionke and Blanchard 1975); the suspended sediment

then has a negative impact on the functioning of hydraulic structures. From this perspective, an accurate estimation of the amount of soil eroded is essential for studies of hydrology, hydraulics, agriculture, and ecosystems (Lee and Lee, 2010).

The assessment of soil erosion and sedimentation requires a basic understanding of the spatial patterns, rates and processes of soil erosion and sediment transport at the watershed scale. However, spatial data are often scarce; thus, our ability to model the spatial patterns of sediment delivery and to identify the source areas of sediment is very limited (Haregeweyn et al., 2013).

When a precipitation event occurs, the eroded soil is transported by a number of routes into local streams (Maidment, 1999). The soil erosion is controlled by the abundance and type of vegetation, the nature of the underlying soil, and the water flow, which eventually produces saturated overland flow. The ratio of sediment yield at the outlet of a basin to soil erosion over the basin is called the sediment delivery ratio (SDR) or the transport capacity coefficient (KTC). The KTC must be determined before sediment production can be estimated. However, this quantity cannot be easily measured. Recent studies have

demonstrated the effects of field boundaries on sediment deposition and the SDR values of different parts of watersheds. These results emphasize the importance of the spatial variability of soil deposition and sedimentation rates that occurs due to the existence of different land use types and the connectivity of watershed characteristics.

To overcome these problems, spatially distributed, process-based models can be used. Several attempts have been made to use such process-based models such as the Water Erosion Prediction Project model (Gete, 1999; Haregeweyn et al., 2013), the

Agricultural Nonpoint Source Pollution model (Haregeweyn and Yohannes, 2003; Hussen et al., 2004) or the Limburg Soil Erosion Model (Hengsdijk et al., 2005). However, such models require large amounts of input data whereas the return in increased accuracy of soil erosion prediction is limited (Jetten et al., 2003). If such models are applied in conditions where the necessary data are not available and/or a proper calibration cannot be performed, the results may become completely unreliable



(Nyssen et al., 2006; Haregeweyn et al., 2013). Spatially distributed empirical or conceptual models may form an alternative to the complex physics-based spatially distributed models. WATEM/SEDEM (Water and Tillage Erosion Model/Sediment Delivery Model) was developed for prediction of sediment yield at the catchment scale with limited data requirements (Van Oost et al., 2000; Van Rompaey et al., 2001). WATEM/SEDEM has been used in various types of environments in (Van Rompaey et al., 2001, 2003, 2005; Verstraeten et al., 2002, 2007), including hydrological catchments in Spain (de Vente et al., 2008; Alatorre et al., 2010).

The KTC values are determined on a grid with a spatial resolution of 1 km by 1 km within the watershed. To overcome the lack of TC field data, the TC values simulated at 181 locations using the Modified USLE (MUSLE) sediment yield algorithm implemented in the Soil and Water Assessment Tool (SWAT) model were applied to determine the spatially distributed KTC values. SWAT has been widely used to evaluate soil erosion and sediment fluxes (Zhu et al., 2008).

The overall goal of this study is to estimate the KTC of the transport capacity (TC) equation in the WATEM/SEDEM algorithm through evaluating the rain erosivity factor R of the Revised Universal Soil Loss Equation (RUSLE) for 14 years (2000~2013) using 1-minute data from 16 rainfall gauging stations in the Han River basin (34,148 km$^2$) of South Korea. The specific objectives of the study are as follows: (1) to develop the TC algorithm and to simulate TC when KTC has a value of 1.0; (2) to calibrate and validate the SWAT model using suspended solids (SS) data to obtain observed TC values, (4) to identify optimal KTC values; and (5) to determine an empirical KTC equation using regression analysis between the estimated KTC values and watershed characteristics and evaluating the usability of the KTC values in the last step. This study assumes that the KTC parameter is dependent on watershed characteristics within the subwatersheds. Therefore, watershed characteristics can be related to KTC.

## 2 Materials and methods

### 2.1 Study area description

The Han River basin (34,148 km$^2$) is one of the five major river basins in South Korea (99,720 km$^2$). It occupies approximately 31 % of the country and falls within a latitudinal range of 36.03° N to 38.55° N and a longitudinal range of 126.24° E to 129.02° E. The basin contains two main rivers, the North Han River (with a drainage basin of 11,342 km$^2$) and the South Han River (12,577 km$^2$), which then merge and flow into the metropolitan city of Seoul, which has 10 million residents. The water resources of the river basin must be managed sustainably, due to the expanding demand for water supply to the capital of Seoul and the surrounding area. The study watershed is divided into 181 subwatersheds, which range in size from 75.4 km$^2$ to 281.5 km$^2$ (Figure 2(a)). This range of sizes corresponds to the standard watershed area considered in the management of water resources and nonpoint source pollution in South Korea. The land use within this watershed is dominated by forest (73.3 %), followed by upland crops (7.9 %), rice paddies (7.3 %), urban areas (5.8 %), and other land uses (5.7 %) (Figure 2(b)). The dominant soil constituents are sand (52.0 %) and loam (26.7 %) (Figure 2(c)). The soil erosion factor K indicates the resistance of soil to erosion by rainfall and is related to the grain size distribution of the soil particles, as well as the structure and organic



matter content of the soil and the sizes of voids within the soil. The values of K typically range from 0.13 to 0.91 Mg/ha/R. In this study, the values of K were estimated using the K factor equation (Lal, 1988; Meusburger et al., 2013) and soil data from the Korea Rural Development Administration (KRDA). In this study, the K values range from 0.1 to 0.9, and the average is 0.41 (Figure 2(d)). The detailed procedure by which the K factor is estimated can be found in Meusburger et al. (2013).

The river basin is in the monsoon area; thus, wet and dry seasons occur each year, producing seasonal variations in precipitation. June through August (i.e., the summer) is usually the wet season, and most of the yearly rainfall occurs during this period. Approximately 30 % of the annual rainfall occurs during the other 9 months. Over the 30 years of weather data from 1985 to 2014, the average annual precipitation was 1,254 mm, and the annual mean temperature was 11.5 °C. However, the amount of rainfall that occurred in summer from 2000 to 2014 was 7.1 % greater than the corresponding summer rainfall

amount for 1985~1999, even though the number of rainy days from June to August decreased from 43 days/year to 37 days/year. The increased rainfall caused stream discharge to increase by 9.9 %, and considerable additional soil erosion and sediment delivery to streams and dam reservoirs occurred.

The suspended solid data as sediment delivery were measured by the Ministry of Environment at 7 locations, Hoengseong Dam (HSD), Soyang Dam (SYD), Chungju Dam (CJD), Paldang Dam (PDD), Kangcheon Weir (KCW), Yeoju Weir (YJW),

and Ipo Weir (IPW) (Figure 1). Sites HSD, SYD, CJD, and PDD have 9 years (2005~2013) of data, whereas sites KCW, YJW, and IPW have 2 years (2012~2013) of data. At or near dam and weir locations, sediment-bearing water is gathered and purified at filtration plants to provide water for municipal and industrial use. Recently, increases in the amount of sediment in the water have led to increased purification costs. Additionally, along the stream, the agricultural water is supplied by water resources of decreased quality that contain larger amounts of sediment, phosphorus, and nitrogen, due to the increased discharge loads

of nonpoint source pollution.



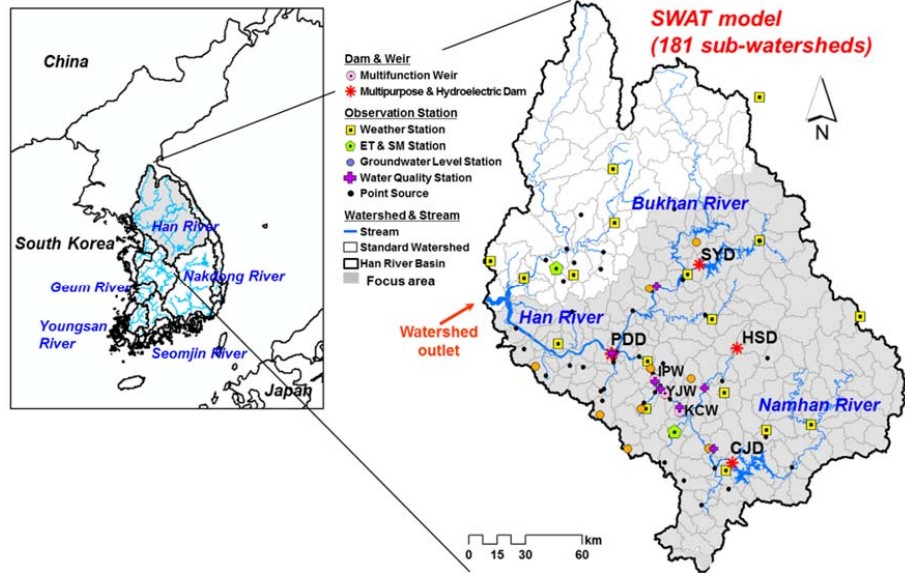

**Figure 1: Locations of the Han River basin (34,148 km²) and gauging stations and study area used in hydrological modeling**

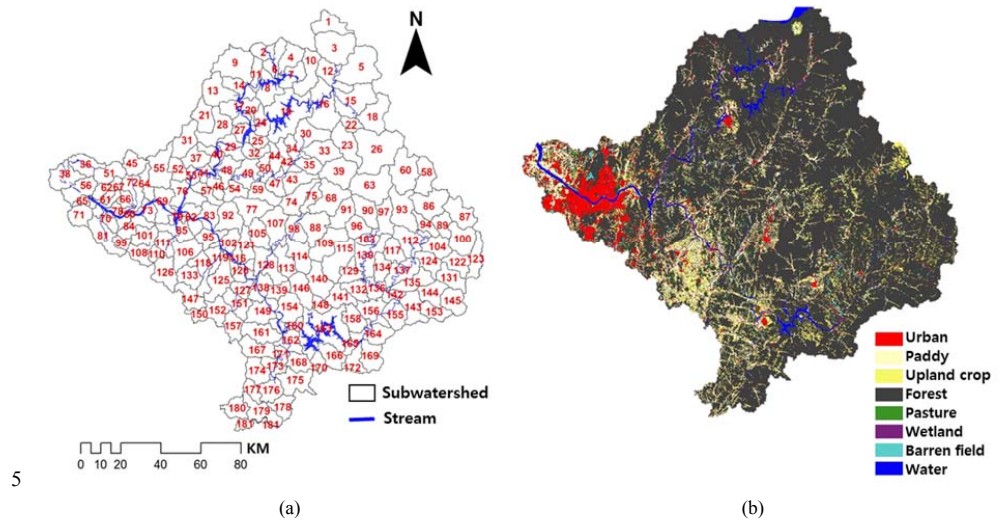

(a)                                                         (b)

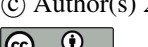



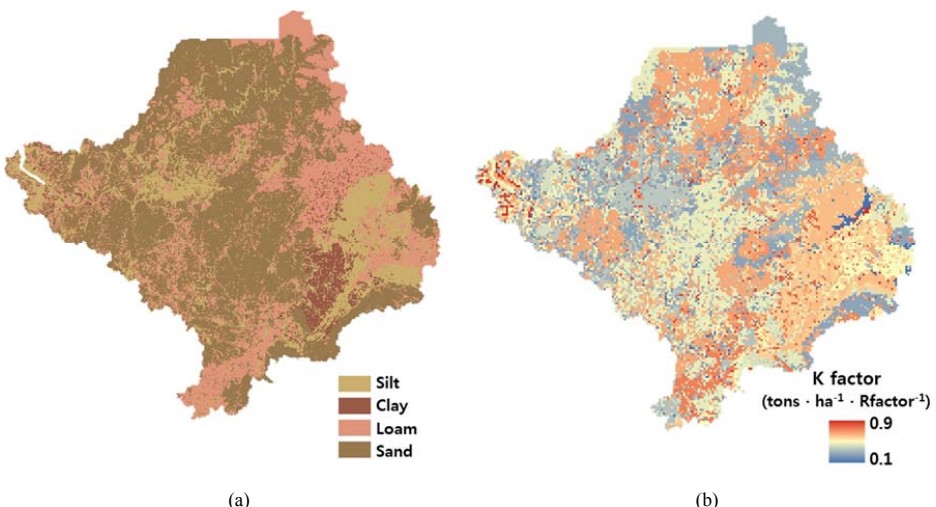

**Figure 2: Distribution maps of (a) subwatershed number, (b) land use, (c) dominant soil constituents, and (d) K factor**

### 2.2 TC equation in WATEM/SEDEM

We use the TC equation in the WATEM/SEDEM algorithm to estimate soil erosion and sediment fluxes to the stream network. The TC equation is a sediment delivery model that is based on the RUSLE model and predicts how much sediment is transported to river channels on an annual basis. It is a spatially semi-distributed model, which means that the landscape is divided into small spatial units or grid cells (Van Oost et al., 2000; Van Rompaey et al., 2001; Verstraeten et al., 2002). The TC equation is as follows:

$$TC = KTC \cdot R \cdot K \cdot (LS - 4.1s^{0.8}) \tag{1}$$

where TC is the soil transport capacity (kg/m$^2$/yr), R is the rainfall erosivity factor (MJ·mm/m$^2$·h·yr), LS is a slope-length factor (Desmet and Govers, 1996), s is the slope gradient, and KTC is an empirical transport capacity coefficient that depends on the soil and geomorphic characteristics of the watershed being considered. To simulate the distribution of sediment delivery, the original algorithm must be modified to produce a fully distributed model.

The mass balance approach used in the distributed model is followed for determining the net amount of sediment in each cell. The sediment transported to each cell from the neighboring upslope cells is added to the sediment generated in the cell by erosion, and this amount is then exported entirely to the downslope cells.

### 2.3 SWAT model

SWAT is a physically based, continuous, long-term, distributed-parameter model designed to predict the effects of land
management practices on hydrology and water quality in agricultural watersheds under varying soil, land use, and management



conditions (Arnold et al., 1998). SWAT is based on the concept of hydrologic response units (HRUs), which are portions of a sub-basin with unique land use, management, and soil attributes. The runoff, sediment, and nutrient loadings from each HRU are calculated separately based on weather, soil properties, topography, vegetation, and land management and are then summed to determine the total loading from the sub-basin (Neitsch et al., 2011; Park et al., 2011; Park et al., 2014). The hydrologic

cycle, as simulated by SWAT, is based on the water balance equation:

$$SW_t = SW_0 + \sum_{i=1}^{t}(R_{day} - Q_{surf} - E_a - W_{seep} - Q_{gw}) \qquad (2)$$

where $SW_t$ is the final soil water content (mm), $SW_0$ is the initial soil water content on day i (mm), t is the time (days), $R_{day}$ is the amounts of precipitation on day i (mm), $Q_{surf}$ is the amount of surface runoff on day i (mm), $E_a$ is the amount of evapotranspiration on day i (mm), $W_{seep}$ is the amount of water entering the vadose zone from the soil profile on day i (mm),

and $Q_{gw}$ is the amount of return flow on day i (mm).

The SWAT model estimates erosion and sediment yield for each HRU using the Modified Universal Soil Loss Equation (MUSLE). This equation includes the detachment of particles by rainfall and flow, the transport of particles by flow and the deposition of particles based on flow power and particle size, which affect the tendency of flowing water to continue to pick up particles or to deposit them. The MUSLE (Williams, 1975; Neitsch et al. 2010) is considered in SWAT to estimate the

erosion produced by both rainfall and surface runoff flow for individual rainfall events using the following equation:

$$Sed = 11.8 \times \left(Q_{surf} \times q_{peak} \times area_{hru}\right)^{0.56} \times K_{usle} \times C_{usle} \times P_{usle} \times LS_{usle} \times CRFG \qquad (3)$$

where $Q_{surf}$ is the surface runoff volume (mm/ha), $q_{peak}$ is the peak runoff rate (m³/s), $area_{hru}$ is the hydrologic response unit area (ha), $K_{usle}$ is the soil erodibility factor of the USLE, $C_{usle}$ is the cover management factor of the USLE, $P_{usle}$ is the USLE support practice factor, $LS_{usle}$ is the USLE topographic factor, and $CRFG$ is the coarse fragment factor.

**2.4 Multiple linear regression model**

Regression analysis is commonly used to measure the relationship between two or more variables and to predict the behavior of a dependent or endogenous variable according to one or more independent or explanatory variables. Multiple linear regression (MLR) models are frequently used as empirical models or approximating functions and to establish mathematical models to describe real-world phenomena. The general relationship between the dependent variables and an independent

variable is presented by Eq. (4) (Prieto et al., 2017).

$$Y = C + \beta_1 X_1 + \beta_2 X_2 + \beta_3 X_3 + \beta_4 X_4 + \ldots + \beta_n X_n \qquad (4)$$

where Y is the dependent variable; C is a constant; $X_1, X_2, X_3, X_4,$ and $X_n$ are independent variables; and $\beta_1, \beta_2, \beta_3, \beta_4,$ and $\beta_n$ are regression coefficients.





## 2.5 Model implementation

To estimate the spatial distribution of sediment delivery, the KTC of the TC equation has to be estimated from field observations of TC. Because it is difficult to observe the KTC directly over large areas, the KTC value is generally set to values ranging from 0.01 to 100 that were originally acquired from specific experimental sites (Van Rompaey et al., 2001,

2003, 2005). The notable scientific accomplishments of this study involve the implementation of the TC model and the subsequent derivation of a relationship between the KTC values and the watershed characteristics.

The derivation of KTC was performed using the existing SDR method. The empirical SDR area function is widely used to estimate SDR. Traditional methods of estimating SDR are also often data driven. These methods depend on the availability of records of sediment yield at stream gauging stations over long periods of time and measurements or estimates of hillslope

erosion rates. However, few sediment yield datasets that are continuous over long periods are available from large regional basins to allow such analyses to be carried out. As drainage area increases, the responses of catchments to changes upstream are often longer than the record length. Other methods that attempt to build models based on fundamental hydrologic and hydraulic processes also exist. In the majority of these models, sediment yield and deposition are predicted through a coupling between runoff and erosion/deposition, conditioned on sediment TC (Flanagan et al., 1995; Lu et al., 2006). Therefore, KTC

values were estimated using suspended solid (SS) data observed at 7 stations and the SWAT hydrologic model. Since KTC is difficult to observe directly, we employ the observed sediment delivery in the calibration and validation of the SWAT model in this study.

For a given catchment area, each individual empirical TC only produces a single value of KTC. In this study, to determine the distributed KTC within a large watershed, the TC values at 181 locations are generated using the SWAT model and TC

field data collected at the 7 locations (HSD, SYD, CJD, PDD, KCW, YJW, and IPW). Thus, the SWAT model divides the watershed as a whole into 181 subwatersheds. Figure 3 shows a schematic diagram of this study. Using the 181 TC values determined for each subwatershed, the 181 KTC values from the TC equation are evaluated. Thus, the KTC within a subwatershed has just one value of KTC, which is an important limitation of the representation of KTC within subwatersheds. If detailed KTC distributions are necessary, the SWAT model requires a greater number of small-scale subwatersheds than

this study examines. The SWAT sediment yield is equal to TC ($kg/m^2/yr$) in the WATEM/SEDEM TC, considering the delivery ratio in the subwatersheds.





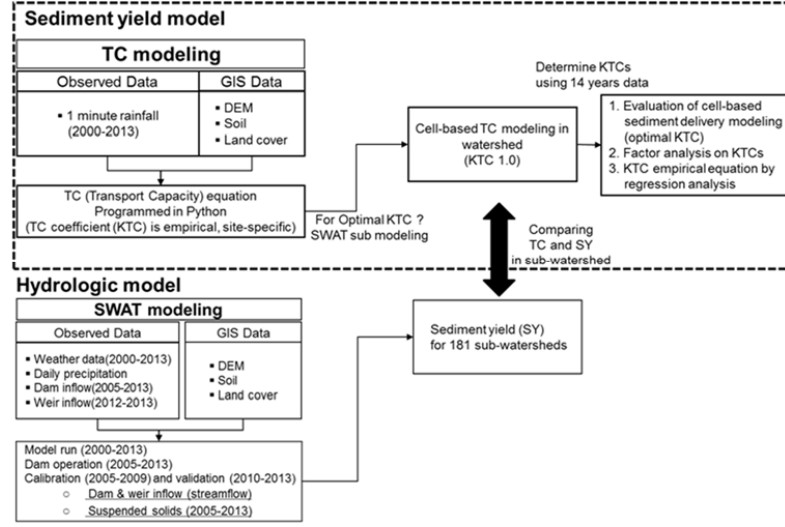

**Figure 3: Flow chart of this study**

## 3 Results and discussion

### 3.1 1-minute R factor estimates

R factor represents the erosivity of the climate at a particular location. An average annual value of R is determined from historical weather records and is the average annual sum of the erosivity of individual storms. The erosivity of an individual storm is computed as the product of the storm's total energy, which is closely related to the amount of water and the storm's maximum 30-minute intensity (Lee, 2004). In this study, an average annual value of R is calculated using 1-minute precipitation data measured at 51 meteorological stations in South Korea. For each storm, the equations developed by

Wischmier and Smith (1978) for calculating R are as follows:

$$e = 210.3 + 89 \log(I) \tag{5}$$

$$E = \sum(e \cdot I \cdot t) \tag{6}$$

$$R = \sum(E \cdot I_{30}) / 100 \tag{7}$$

where R is the rainfall erosivity factor (MJ·mm·ha$^{-1}$·h$^{-1}$·y$^{-1}$), $E$ is the total kinematic energy of the storm, $I_{30}$ is the maximum

30-minute rainfall intensity, $e$ is the kinematic energy of an individual storm event (metric·tons·m$^{-1}$·ha$^{-1}$·cm$^{-1}$), $I$ is the rainfall intensity (cm·hr$^{-1}$), and $t$ is the duration of the rainfall event (hr). A detailed description of the procedure used to estimate R is



found in Lee, (2004). The result, in the form of a rain erosivity map, is shown in Figure 4. The yearly distributed R factor of the TC equation is evaluated using 1-minute rainfall data with 1 km ×1 km grid cells. The model estimates an average R factor of 3,812 MJ/ha mm/year during 2000–2013 over the study area as a whole (Figure 4). Additionally, the range of maximum R factor values during 2000–2013 is 1438.6–13616.1 MJ/ha mm/year.

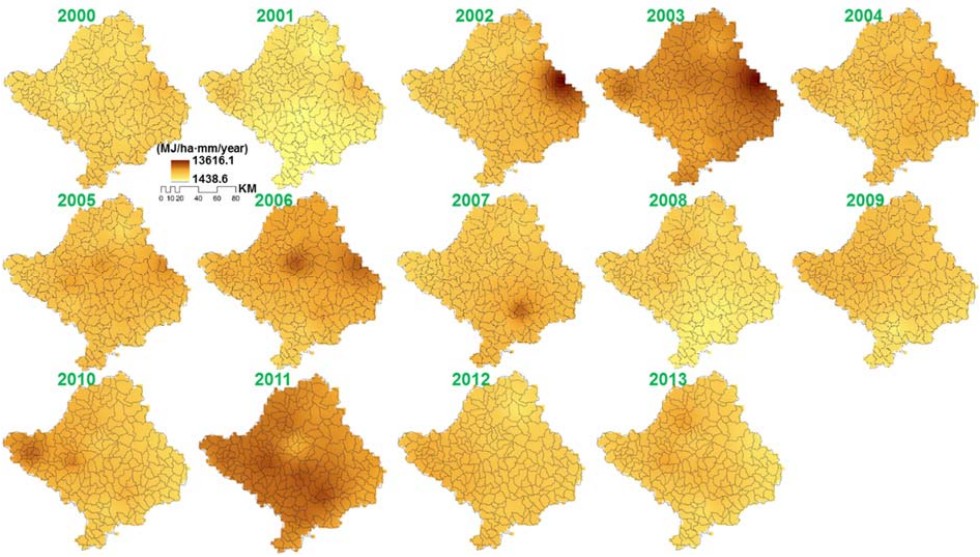

**Figure 4: The distribution of the rain erosivity factor using 1-minute rainfall data from 2000–2013**

### 3.2 1 km ×1 km TC calculations performed by setting KTC to 1.0 using a 1 minute R estimates

The yearly distributed sediment delivery (kg/m$^2$/year) based on the TC equation is evaluated using a KTC value of 1.0 as an
10  initial condition with 1 km ×1 km grid cells, corresponding to the subwatershed scale. Afterwards, the 1.0 KTC value is adjusted by comparing the sediment delivery obtained using the TC equation with the TC simulated using the SWAT model on the subwatershed scale.

The model predicts an average sediment delivery (SD) of 0.134 kg/m$^2$/year during 2000–2013 in the overall study area (Figure 5). Additionally, the range of maximum SD values during 2000–2013 is 0.954–2.711 kg/m$^2$/year. The regions of high
15  sediment delivery can be explained in terms of the K factor, high values of the LS factor and comparatively abundant highland agricultural land compared to other regions.





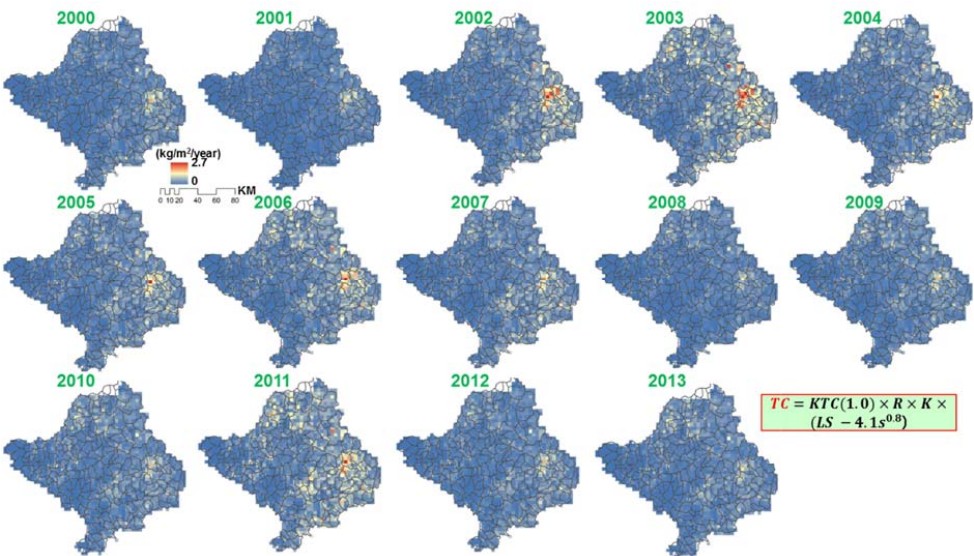

**Figure 5: The distribution of sediment delivery predicted by TC modeling (KTC = 1.0) from 2000–2013 year**

### 3.3 Calibration and validation of the SWAT model

The SWAT model simulates hydrological components and water quality, including suspended solids, related to TC. The
SWAT model is calibrated at seven locations (HSD, SYD, CJD, KCW, YJW, IPW, and PDD) in the main river reaches using
five years (2005–2009) of daily inflow (streamflow) data to the dams and weirs. The SWAT model is subsequently validated
using another four years (2010–2013) of data and the average calibrated parameters (Figure 6). As seen in Figure 7, the model
is spatially calibrated and validated using evapotranspiration and soil moisture data measured at two locations over five years
(2009–2013). In the case of dam inflow, the R² value is greater than 0.59. The average Nash-Sutcliffe efficiency (NSE) at
HSD, SYD, CJD, KCW, YJW, IPW, and PDD is 0.59, 0.78, 0.61, 0.79, 0.77, 0.88, and 0.87, respectively, whereas the
corresponding percent bias (PBIAS) values are 13.5 %, 12.2 %, 9.4 %, 11.5 %, 19.8 %, 21.4 %, and 4.5 %, respectively. In
the case of dam storage volume, the average R² values are between 0.40 and 0.96, and the PBIAS is between 0.9 % and 18.9 %
for each calibration point. The average R² values for evapotranspiration are between 0.77 and 0.72, whereas those of soil
moisture and groundwater level are between 0.80 and 0.78 and between 0.47 and 0.68, respectively, for each calibration point
(Table 1 and Table 2). For a detailed description of the results, see the paper by Ahn and Kim (2016, under review). When
checked against the guidelines for SWAT calibration (NSE ≥0.5, PBIAS ≤28 %, and R² ≥0.6, Moriasi et al., 2007), the results
are found to be satisfactory.

In this study, the SWAT model is used to simulate SS within the Han River basin. The SWAT model is calibrated at the
same locations where streamflow was measured in the main river reaches using five years (2005–2009) of SS data collected





at eight-day intervals and subsequently validated using another three years (2010–2013) of data using the average calibrated parameters (Figure 8). The average $R^2$ for SS is between 0.61 and 0.80 for each calibration point. The average $R^2$ for streamflow is typically greater than 0.60, which indicates satisfactory simulation performance.

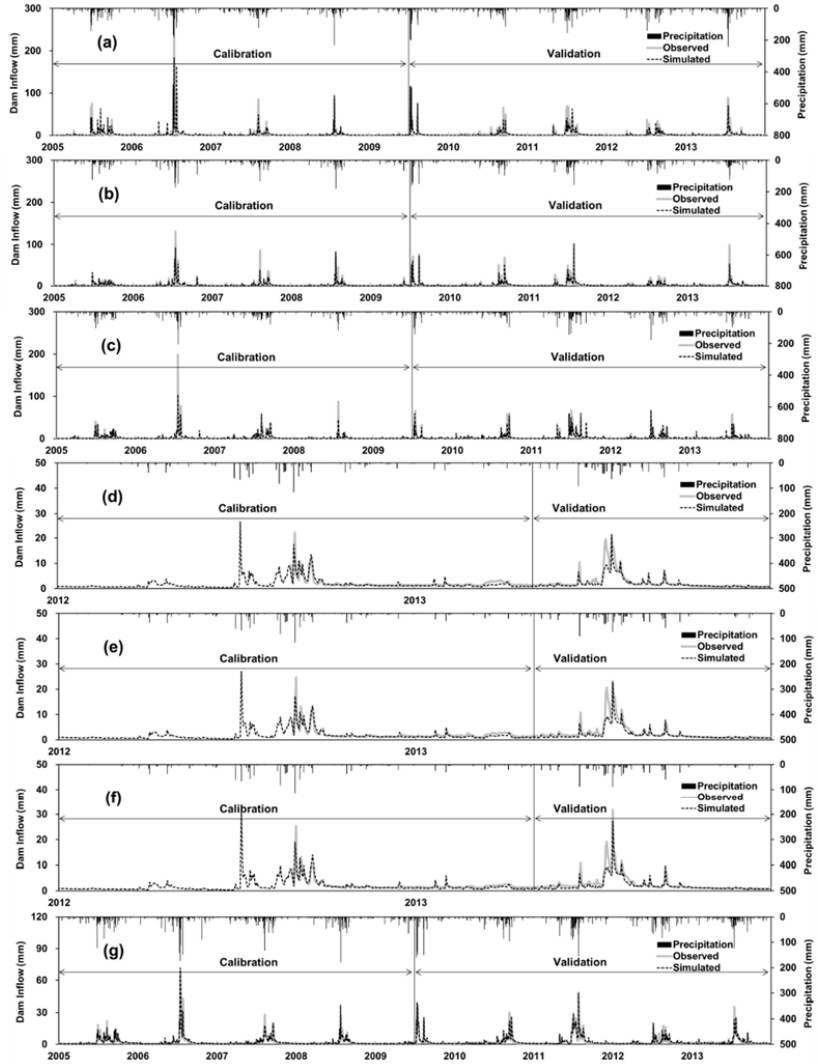





**Figure 6: Comparison of the observed and SWAT-simulated daily dam inflow (streamflow) during the calibration (2005–2009) and validation (2010–2013) periods at (a) HSD, (b) SYD, (c) CJD, (d) KCW, (e) YJW, (f) IPW, and (g) PDD**

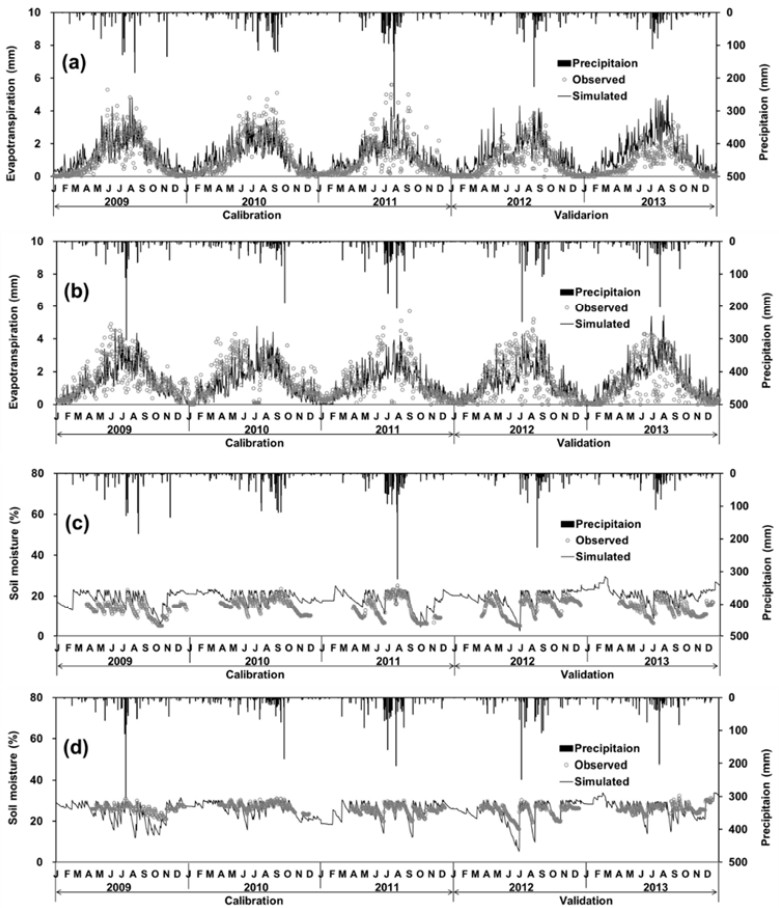

5     **Figure 7: Comparison of the observed and SWAT-simulated daily evapotranspiration at (a) SM and (b) CM and soil moisture at (c) SM and (d) CM from 2009–2013**




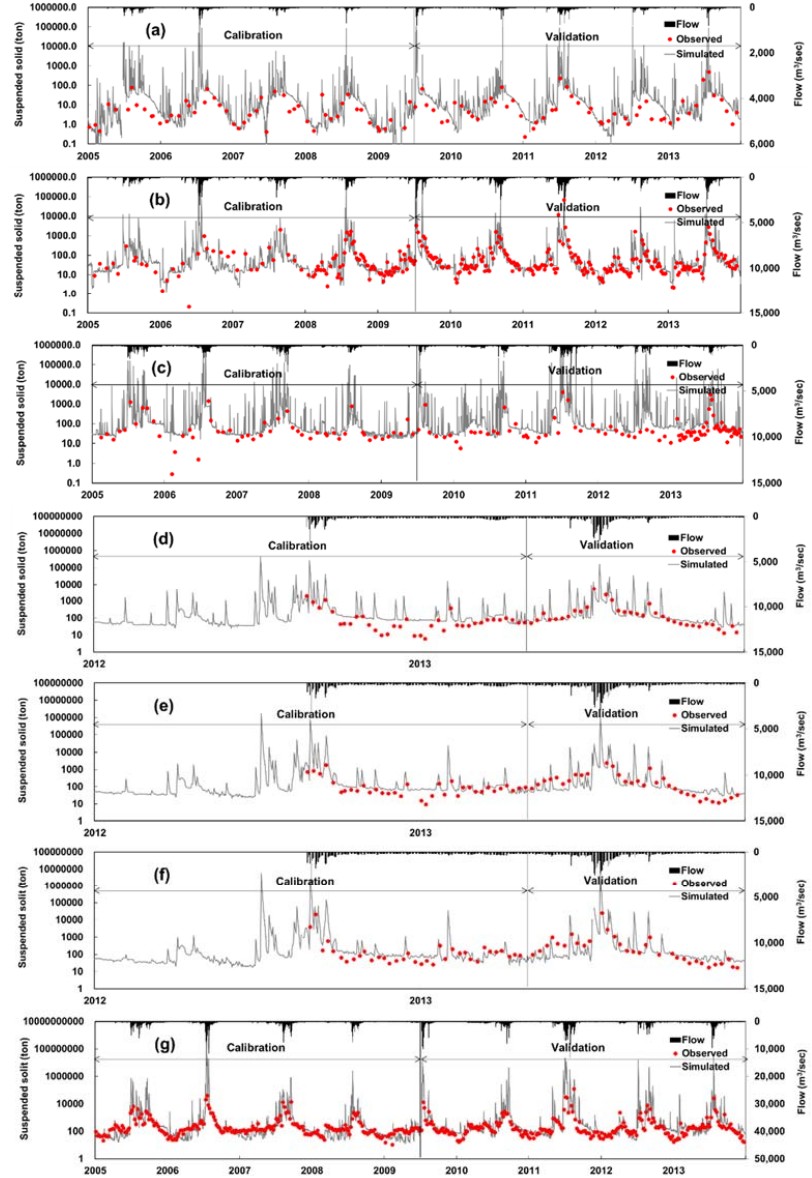

**Figure 8: Comparison of the observed and SWAT-simulated daily sediments during the calibration (2005–2009) and validation (2010–2013) periods at (a) HSD, (b) SYD, (c) CJD, (d) KCW, (e) YJW, (f) IPW, and (c) PDD**

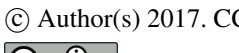


**Table 1.** Calibration and validation results for dam inflow and storage at seven calibration points in the main reach

| Model output | Evaluation criteria | HSD | | SYD | | CJD | | KCW | | YJW | | IPW | | PDD | |
|---|---|---|---|---|---|---|---|---|---|---|---|---|---|---|---|
| | | Cal. | Val. | Cal. | Val. | Cal. | Val. | Cal. | Val. | Cal. | Val. | Cal. | Val. | Cal. | Val. |
| Dam inflow (mm) | R² | 0.82 | 0.84 | 0.90 | 0.89 | 0.81 | 0.74 | 0.90 | 0.63 | 0.91 | 0.62 | 0.93 | 0.59 | 0.92 | 0.88 |
| | NSE | 0.61 | 0.57 | 0.78 | 0.78 | 0.63 | 0.58 | 0.78 | 0.79 | 0.77 | 0.76 | 0.81 | 0.95 | 0.83 | 0.76 |
| | RMSE (mm/day) | 7.9 | 9.3 | 3.8 | 3.9 | 3.5 | 3.1 | 6.5 | 0.7 | 9.1 | 2.4 | 9.2 | 2.9 | 0.8 | 2.3 |
| | PBIAS (%) | 14.5 | 12.5 | 10.3 | 14.0 | 8.9 | 9.9 | 18.0 | 4.9 | 25.5 | 14.1 | 25.6 | 17.2 | 2.2 | 6.8 |

Cal. = calibration period (HSD, SYD, CJD and PDD: 2005-2009; KCW, YJW and IPW: 2013) and Val. = validation period (HSD, SYD, CJD and PDD: 2010-2014; KCW, YJW and IPW: 2014)

**Table 2.** Calibration and validation results for evapotranspiration and soil moisture at two calibration points and groundwater level fluctuations at five calibration points in the watershed

| Model output | Evaluation criteria | SM | | CM | | GPGP | | YPGG | | YPYD | | YIMP | | HCGD | |
|---|---|---|---|---|---|---|---|---|---|---|---|---|---|---|---|
| | | Cal. | Val. | Cal. | Val. | Cal. | Val. | Cal. | Val. | Cal. | Val. | Cal. | Val. | Cal. | Val. |
| Evapotranspiration (mm) | R² | 0.81 | 0.73 | 0.70 | 0.74 | - | - | - | - | - | - | - | - | - | - |
| | NSE | 0.64 | 0.45 | 0.50 | 0.55 | - | - | - | - | - | - | - | - | - | - |
| | RMSE (mm/day) | 2.3 | 9.1 | 4.0 | 3.0 | - | - | - | - | - | - | - | - | - | - |
| | PBIAS (%) | 9.6 | 30.2 | 11.6 | 23.7 | - | - | - | - | - | - | - | - | - | - |
| Soil moisture (%) | R² | 0.85 | 0.75 | 0.78 | 0.78 | - | - | - | - | - | - | - | - | - | - |
| Groundwater level (El. m) | R² | - | - | - | - | 0.70 | 0.63 | 0.64 | 0.45 | 0.70 | 0.41 | 0.53 | 0.40 | 0.69 | 0.67 |

Cal. = calibration period (2009-2011) and Val. = validation period (2012-2013)

### 3.4 The determination of KTC using SWAT TC values

The scientific insights to be found in this study are the connections between KTC and watershed characteristics. Therefore,
this study assumes that the KTC parameter is dependent on watershed characteristics within the subwatersheds. Based on the TC results obtained using a KTC of 1.0, the KTC in the TC equation is calculated using the SWAT TC results for each subwatershed. The 181 KTC values are calculated for each subwatershed by comparing with the SWAT TC results covering 14 years (2000–2013). The 14 years of KTC values are subsequently averaged for each subwatershed. The average KTC value for the whole watershed is 12.58, and the range of the KTC values for the 181 subwatersheds are 0.16–112.58 (Figure 9).
However, the KTC values should be re-evaluated if the watershed area exceeds 281.5 km².

The KTC distribution (Figure 9(c)) shows that the KTC values in the lower portions of the watershed are relatively high when compared to the values in the upper watershed, consistent with the SWAT TC results (Figure 9(a)). As shown in Table 3, the KTC values increase as the silt and clay increase from the upper (1 to 70) to the lower (111 to 181) portions of the watershed locations. This result reflects the relatively high vulnerability of silt and clay to soil erosion when compared to loam
and sand, based on the delivery potential of different particle sizes.



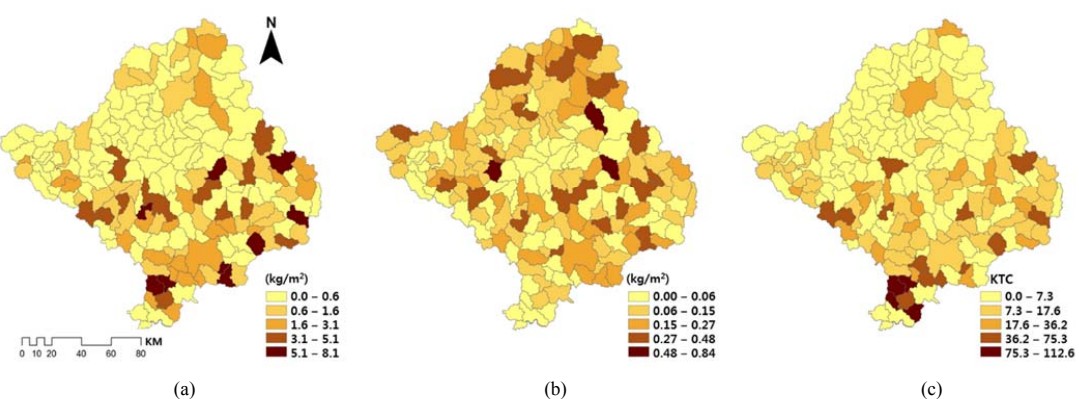

**Figure 9: Maps showing the distribution of (a) SWAT TC values obtained through SWAT modeling (using a KTC value of 1.0), (b) TC values obtained using the TC equation, and (c) estimated KTC values.**

**Table 3.** Percentage of soil type (silt, clay, loam, and sand) according to the locations of different subwatersheds (upper, middle, and lower) within the watershed.

| Subwatershed number in Figure 2(a) | Silt area | | Clay area | | Loam area | | Sand area | |
|---|---|---|---|---|---|---|---|---|
| | (km²) | (%) | (km²) | (%) | (km²) | (%) | (km²) | (%) |
| 1 to 70 (upper watershed) | 227.2 | 5.1 | 7.3 | 0.2 | 1,265.2 | 28.4 | 2,957.2 | 66.4 |
| 71 to 110 (middle watershed) | 995.7 | 14.6 | 108.5 | 1.6 | 1,588.1 | 26.3 | 2,577.4 | 57.5 |
| 111 to 181 (lower watershed) | 2,614.7 | 20.6 | 785.2 | 6.2 | 3,142.1 | 24.7 | 6,173.5 | 48.6 |

### 3.5 Factor-based generalization of KTC (Scenario 1)

In section 3.3, the KTC was calculated for each subwatershed. The KTC varies with different soil texture, land use and
10 watershed characteristics. To generalize the KTC values according to the watershed characteristics, a simple regression equation is generated using selected KTC impact factors, such as watershed slope, watershed area, and the K factor of the MUSLE (Alatorre et al., 2012). The relationship between watershed slope and KTC shows a low Pearson's coefficient of 0.12. The watershed area and K factor show strong relationships, with Pearson's coefficients of -0.68 and 0.72 (Figure 10(a) and (b)). Therefore, the watershed area and K factor were employed to generalize the estimation of KTC values for particular
15 watersheds (Figure 10(c)). Thus, a KTC estimation equation that employs the watershed area and K factor is derived that has a coefficient of determination $R^2$ of 0.63:

KTC = 151.9 × K factor − 0.16 × watershed area (km²)                    (8)



Examination of Figure 10(c) shows that the estimated KTC values produced using Equation (8) tend to overestimate the KTC values estimated using the SWAT TC results. Figure 11 shows the trend of the differences between the three estimated values for subwatersheds 1 to 181, and Table 4 shows a statistical summary of the KTC differences for 8 subwatersheds, 175, 177, 180, and 181 (with high-ranked KTC values within 2 %) and 30, 65, 72, and 78 (with low-ranked KTC values within

2 %). The KTC differences are larger up to 52.1, where the watershed area decreases and the K factor increases. Although $R^2$ values greater than 0.6 are considered to represent good correlations (Donigian et al., 2000), Equation (8) leaves 40 % of the variance unexplained when KTC values are estimated using just these 2 factors.

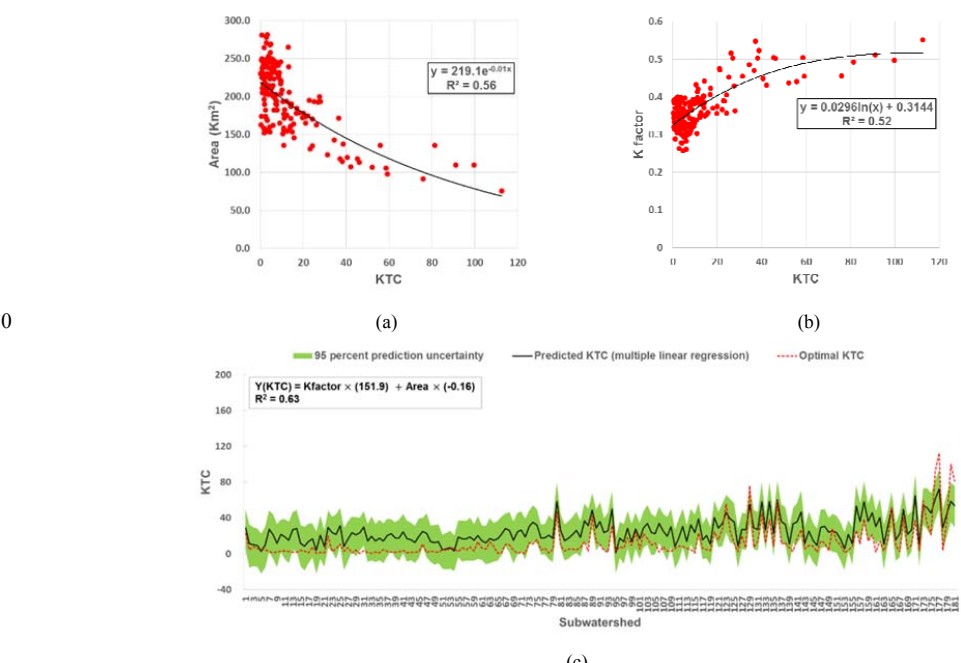

**Figure 10: The results of the KTC regression analysis for Scenario 1: (a) area and KTC, (b) K factor and KTC, and (c) multiple regression analysis between area and the K factor.**





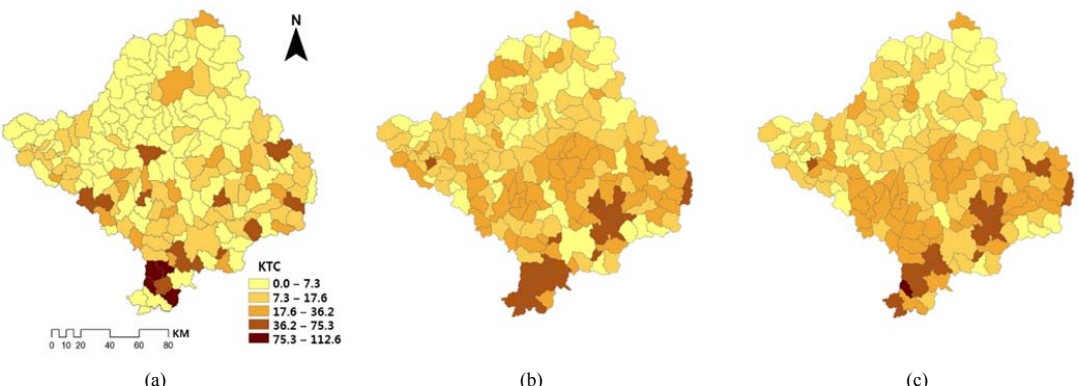

**Figure 11: Distribution maps of KTC in each subwatershed: (a) optimal KTCs, (b) predicted KTCs from the MLR analysis of Scenario 1, and (c) predicted KTCs from the MLR analysis of Scenario 2**

**Table 4.** Statistical summary of the differences in KTC for 8 subwatersheds, 175, 177, 180, and 181 (with high-ranked KTC values within 2 %) and 30, 65, 72, and 78 (with low-ranked KTC values within 2 %).

| Subwatershed number | | Watershed area (km²) | Watershed K factor | Upland crop area (%) | KTC from SWAT results (A) | KTC from equation (5) (B) | KTC from equation (6) (C) | Difference (A-B) | Difference (A-C) |
|---|---|---|---|---|---|---|---|---|---|
| High-ranked KTC | 175 | 109.8 | 0.51 | 29.7 | 91.1 | 49.2 | 71.3 | 41.9 | 19.8 |
| | 177 | 75.4 | 0.55 | 35.5 | 112.6 | 60.8 | 87.2 | 51.8 | 25.3 |
| | 180 | 109.9 | 0.50 | 30.8 | 99.8 | 47.7 | 71.2 | 52.1 | 28.6 |
| | 181 | 135.7 | 0.49 | 29.3 | 81.4 | 42.0 | 65.9 | 39.4 | 15.5 |
| | Mean | 107.7 | 0.51 | 31.3 | 96.2 | 49.9 | 73.9 | 46.3 | 22.3 |
| Low-ranked KTC | 30 | 162.4 | 0.32 | 4.7 | 0.4 | 11.9 | 11.6 | -11.6 | -11.2 |
| | 65 | 229.0 | 0.34 | 2.4 | 0.5 | 5.0 | 4.3 | -4.5 | -3.8 |
| | 72 | 230.2 | 0.36 | 1.9 | 0.2 | 6.7 | 5.0 | -6.6 | -4.9 |
| | 78 | 247.7 | 0.39 | 2.1 | 0.5 | 9.0 | 7.3 | -8.5 | -6.8 |
| | Mean | 217.3 | 0.35 | 2.8 | 0.4 | 8.2 | 7.1 | -7.8 | -6.7 |
| Average of all subwatersheds (1-181) | | 197.9 | 0.37 | 8.4 | 12.6 | 25.2 | 18.7 | -12.6 | -6.1 |

### 3.6 Factor-based generalization of KTC including agricultural land use conditions (Scenario 2)

To reduce the differences between the KTC values estimated from Equation (8) and the KTC values estimated from the results of the SWAT model, the land use factors of upland crop area and rice paddy area within each subwatershed are introduced into the regression analysis. As seen in Figure 12(a) and (b), the upland crop area is positively correlated with the KTC values





(R$^2$ of 0.51). Thus, a new multiple regression equation that includes the percentage of upland crop area within subwatersheds is derived as follows. This MLR equation yields an improved R$^2$ of 0.76, as compared to the 0.63 obtained using Equation (9);

KTC = 117.6 × K factor – 0.11 × watershed area (km$^2$) + 1.26 × upland crop area (%) – 14.03     (9)

Looking at Figure 12(c), the KTC values from Equation (9) reflect improved estimates compared to the KTC values obtained using Equation (8). As seen in Table 4, the KTC differences from Equation (9) decrease from 46.3 to 22.3 for the 4 subwatersheds with the 2 % high-ranked KTC values and from -12.6 to -6.1 for the 4 subwatersheds with the 2 % low-ranked KTC values.

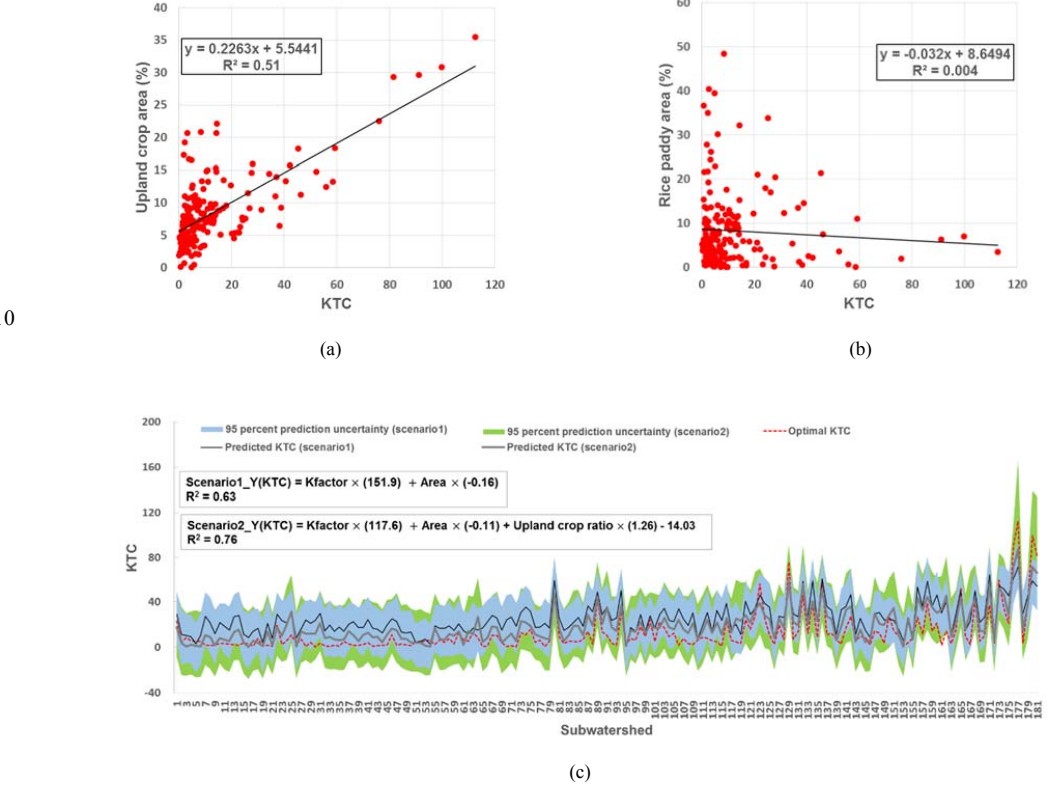

(a)                          (b)

(c)

**Figure 12: The results of the KTC regression analysis for Scenario 1 and Scenario 2: (a) ratio of upland crop area and KTC, (b) ratio of rice paddy area and KTC, and (c) MLR analysis between watershed area, K factor, and ratio of upland crop area.**





**4 Summary and conclusion**

We attempted to determine the spatial variations of the KTC coefficient in the WATEM/SEDEM sediment delivery TC equation using the SWAT sediment yield estimated for each subwatershed unit and to derive a generalized equation based on watershed characteristics.

The TC sediment yields obtained from the SWAT MUSLE results for 181 locations were used to estimate the KTC values of the 181 subwatersheds. The KTC values obtained using the TC equation were 12.58, on average, and these values displayed a wide range that extended from 0.16 to 112.58. To determine the KTC without employing the SWAT model, the watershed area, MUSLE K, and the upland crop area (%) were used, resulting in an $R^2$ value of 0.76. Although $R^2$ values greater than 0.6 are usually classified as reflecting a good fit, the deviation of the predicted KTC values became greater for smaller values of

watershed area and upland crop area and higher values of the K factor for individual watersheds. In cases where a watershed is divided into subwatersheds with the same area, the derived KTC equation can be used with just 2 factors, land use and soil information, and can help to establish a practical strategy for the mitigation of soil erosion due to upland crop practices and new methods of upland crop cultivation.

    Future research could be improved if the soil classification were to be further subdivided to the 12 major textural classes

and sediment production were simulated based on larger amounts of observed data. Additionally, it is necessary to understand the changes in KTC directly in terms of various conditions that affect the K factor, such as forests and urban density.

**Acknowledgments**

   This research was supported by a grant (17AWMP-B079625-04) from the Water Management Research Program of the Ministry of Land, Infrastructure and Transport of the Korean government.

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
