# Peer review of "Empirical Estimation of the Spatial Sediment Transport Capacity Coefficient Using the Rain Erosivity Factor and SWAT Model Results in the Han River Basin, South Korea"

_Hydrology and Earth System Sciences, 2017_

## Referee Comment (RC1) · Anonymous Referee #1 · 20 Oct 2017

The authors submitted the revised version of the manuscript as a new submission (first submission had manuscript number hess-2016-649). The manuscript is slightly improved. However, the main drawbacks from my point of view are written below and are similar to the ones stated in the review process of the first submission:

-The structure of the paper is still not well organized (e.g., some paragraphs should be moved from the Results and discussion section to the methodology, for example eq. 8 and eq. 9); it is still hard to understand the individual parts of the research work that has been done and consequently the paper messages are not clear.

[Figure]

-Some steps of the research are still not well described, e.g., how suspended sediments were measured (eight days interval is mentioned but not explained), how was SWAT model calibrated (which method was used), etc. (more suggestions were given in the review of the first submission).

-There are a lot of small errors, e.g., the one related to the K factor units, on page 4: sometimes K factor does not have units; sometimes Mg/ha/R units are used? K factor units are usually t ha h ha-1 MJ-1 mm-1. What is the physical meaning of negative KTC values that can be seen as potential result from Fig. 12 and Fig. 10 (95 % confidence interval)?

To sum up, I suggest to reject the paper in its current form and I suggest authors to firstly take into consideration all the comments that were given in the review of the first submission since most of the issues have not been resolved.

---

## Referee Comment (RC2) · Anonymous Referee #2 · 28 Oct 2017

Major Comments:

This study largely focused on estimating spatial sediment transport capacity coefficient (KTC) of TC equation in WATEM/SEDEM algorithm for sediment delivery. The manuscript is interesting and of interest to a broad HESS audience. However, my review consists of a couple of critical issues, which I consider mandatory to be properly addressed in the revised version.

1. This manuscript lacks detailed explanations as to why this study was conducted and

how and where it was applied in both abstract and introduction sections. Thus, please provide the descriptions for "why did this study begin?" and "what is this study pursuing?"; the objectives of the research are not very well pointed out in these sections.

2. The novelty of the paper is scarce. What kind of scientific contribution exist in this manuscript with sediment delivery ratio? The cited references are also not the most recent studies.

3. It is not clear that the main target for estimation; confused with different models (i.e., RUSLE, MUSLE, WATEM/SEDEM, SWAT) and their factors. I think the sediment transport capacity coefficient (KTC) of WATEM/SEDEM is the main factor the author to estimate, but the rain erosivity (R) factor of RUSLE is mainly described in the abstract. Thus, the abstract need to be rewritten underlining the clear objective as well as novelty of the paper.

4. In introduction section, there are some no clear sentences that need to be rewritten. For example, there is no description why the rain erosivity factor R of RUSLE is evaluated for the KTC estimation.

5. In materials and method section, the soil erosion factor K and the suspended solid data in the study area description are rather related to the data and therefore need to be described in a separate sub-section. Also, there is no description of the data used in the overall study, so please write them (e.g. R in section 3.1.1, LS, s, SWAT inputs, etc.) in a separate sub-section.

6. In results and discussion section, the resulted factor-based generalization of KTC in scenario 1 and 2 should be described in the previous section; why did the author draw these different results?

7. In summary and conclusion section, there are some no clear sentences that need to be rewritten. Please see lines 7 to 10; is there a prior design (to determine the KTC without employing the SWAT model) to derive these results?

Minor Comments:

I do not fully provide minor comments on the current manuscript since I think that the paper will require extensive re-write prior to publication.

1. Page 3 lines 2 to 6: Recent studies (2010 to present) for WATEM/SEDEM should be added in this paragraph.

2. Page 3 line 15: Typographic error can be found in this line.

3. Figure 7 and Table 2: Please check the use of CM and SM, what does CM and SM represent?